# Characterization of CD4-Positive Lymphocytes in the Antiviral Response of Olive Flounder (*Paralichthys oliveceus*) to Nervous Necrosis Virus

**DOI:** 10.3390/ijms21114180

**Published:** 2020-06-11

**Authors:** Jae Wook Jung, Jin Hong Chun, Jung Seok Lee, Si Won Kim, Ae Rin Lee, Jaesung Kim, Jassy Mary S. Lazarte, Young Rim Kim, Hyoung Jun Kim, Kim D. Thompson, Tae Sung Jung

**Affiliations:** 1Laboratory of Aquatic Animal Diseases, Research Institute of Natural Science, College of Veterinary Medicine, Gyeongsang National University, 501 Jinju, Gyeongnam 52828, Korea; wjdwodnr0605@gmail.com (J.W.J.); hilanamang@naver.com (J.H.C.); leejs058@gmail.com (J.S.L.); ksw0017@hanmail.net (S.W.K.); gladofls@naver.com (A.R.L.); afteru70@gmail.com (J.K.); jassylazarte@yahoo.com (J.M.S.L.); yl0808@nate.com (Y.R.K.); 2Laboratory of Aquatic Animal Quarantine, General Service Division, National Fishery Products Quality Management Service, Busan 49111, Korea; hjkim1882@korea.kr; 3Moredun Research Institute, Pentlands Science Park, Bush Loan, Penicuik, Midlothian EH26 0PZ, UK; kim.thompson@moredun.ac.uk; 4Centre for Marine Bioproducts Development, College of Medicine and Public Health, Flinders University, Bedford Park, Adelaide, SA 5042, Australia

**Keywords:** CD4 T lymphocytes, helper T cells, monoclonal antibody, nervous necrosis virus, olive flounder

## Abstract

The presence of CD4 T lymphocytes has been described for several teleost species, while many of the main T cell subsets have not been characterized at a cellular level, because of a lack of suitable tools for their identification, e.g., monoclonal antibodies (mAbs) against cell markers. We previously described the tissue distribution and immune response related to CD3ε and CD4-1 T cells in olive flounder (*Paralichthys oliveceus*) in response to a viral infection. In the present study, we successfully produce an mAb against CD4-2 T lymphocytes from olive flounder and confirmed its specificity using immuno-blotting, immunofluorescence staining, flow cytometry analysis and reverse transcription polymerase chain reaction (RT-PCR). Using these mAbs, we were able to demonstrate that the CD3ε T cell populations contain both types of CD4^+^ cells, with the majority of the CD4 T cell subpopulations being CD4-1^+^/CD4-2^+^ cells, determined using two-color flow cytometry analysis. We also examined the functional activity of the CD4-1 and CD4-2 cells in vivo in response to a viral infection, with the numbers of both types of CD4 T cells increasing significantly during the virus infection. Collectively, these findings suggest that the CD4 T lymphocytes in olive flounder are equivalent to the helper T cells in mammals in terms of their properties and function, and it is the CD4-2 T lymphocytes rather than the CD4-1 T cells that play an important role in the Th1 immune response against viral infections in olive flounder.

## 1. Introduction

In vertebrates, the adaptive immune system is mostly mediated by the two major subsets of T lymphocytes, helper and cytotoxic T cells, defined by their expression of CD4 and CD8 glycoproteins, respectively [1,2]. Mammalian CD4-positive T cells regulate the immune responses of the host defense [3]. The CD4 molecules, which are involved in T cell development and activation, are significant cell surface markers used for identifying helper T cell subsets [4,5]. CD4 helper T cells recognize exogenous pathogens through their interaction with the major histocompatibility complex (MHC) class II molecules expressed on antigen presenting cells (APCs), such as dendritic cells and macrophages [1]. Identification of their function in teleost fish has suggested that fish CD4 helper T cells possess a similar role to mammalian helper T cells.

In teleosts, two types of CD4-like molecules, CD4-1 and CD4-2, have been reported in fugu (*Takifugu rubripes*), rainbow trout (*Oncorhynchus mykiss*), carp (*Cyprius carpio*), sea bass (*Dicentrarchus labrax*) and Atlantic salmon (*Salmo salar*) [6,7,8,9,10,11]. Structurally, mammalian CD4 is a membrane glycoprotein with an estimated molecular mass of ~55 kDa that is composed of four immunoglobulin (Ig)-like extracellular domains (D1–D4), a transmembrane domain and a cytoplasmic tail [1,11,12]. Teleost CD4, referred to as CD4-1, is present similar to mammalian CD4 molecules, whereas the second CD4 homologue, designated CD4-2, contains two or three Ig-like domains [6,7,8,11]. Both CD4-1 and CD4-2 molecules possess a cytoplasmic domain with a p56lck tyrosine kinase motif, and these CD4 molecules appear to be equivalent to those in mammals despite low sequence identities [13]. However, there are no detailed studies investigating the differences in the distribution or roles of CD4-1 and CD4-2 lymphocytes in teleosts.

Monoclonal antibodies (mAbs) against cell surface markers are one of the most frequently used tools for characterizing mammalian leukocytes, and these are also used for a variety of research and therapeutic applications [1,14]. Although the number of studies characterizing components of the immune response of teleosts is increasing, studies investigating T cell populations and their function are hindered by the lack of specific mAbs for characterizing target T cells [11,14,15,16]. Many of the mAbs that have been developed against cell markers of teleost leukocytes so far focus on B cells, granulocytes and thrombocytes. While antibodies specific to CD4 lymphocytes are also present in teleost [3,16], including olive flounder [1,17,18,19], further studies have not been implemented due to the deficiency of the mAbs detecting only CD4-2 lymphocyte, one of the CD4 lymphocytes. The aim of the present study was to produce an mAb against CD4-2 lymphocytes of olive flounder (*Paralichthys olivaceus*). This is an economically important aquaculture species in South Korea, and is used as a model fish species for studying the immune system of teleosts. We then used this mAb to examine the distribution of CD4-2-positive cells in different tissues and the distribution of different CD4 lymphocyte populations during a nervous necrosis virus (NNV) infection in olive flounder, a virus capable of causing severe mortality in many different fish species.

## 2. Results

### 2.1. Specificity of Anti-CD4-2 mAb (3C8)

The specificity of the mAb was confirmed by Western blot analysis under reducing conditions and, as expected, the anti-CD4-2 mAb (3C8) detected different amounts of recombinant CD4-2 (rCD4-2) protein in the Western blot, ranging from 16 ng to 500 ng, corresponding with a band of the estimated molecular weight for the protein of ~23 kDa (Figure 1A). The mAb (3C8) did not exhibit any reactivity with rCD3ε or rCD4-1 (Figure 1A). It was established that mAb (3C8) could detect the whole CD4-2 protein, by the screening HEK 293F cells transfected with the whole CD4-2 sequence with the mAb using confocal imaging and flow cytometric analysis. Immunofluorescent staining of CD4-2-positive HEK 293F cells, with the anti-CD4-2 mAb (3C8), showed that the CD4-2 molecule was located on the surface of the cells, while no fluorescence staining was seen in the negative control (Figure 1B). Compared with the populations of cells expressing the CD4-2 molecule, a number of cells (~54%) exhibited the surface expression reacted by anti-CD4-2 mAb (3C8), and only ~1% of the negative controls (Figure 1B). Moreover, when leukocytes from the spleen, head-kidney and peripheral blood of olive flounder were subjected to Western blot analysis, the bands recognized by the anti-CD4-2 mAb (3C8) were detected at ~23 kDa (Figure 1C), consistent with the Western blotting results with the recombinant protein in Figure 1A. A proportion of leukocytes from olive flounder head-kidney were recognized by mAb 3C8, confirmed using fluorescence microscopy. No fluorescent staining could be observed in negative control cells, treated with only FITC. In the mAb 3C8-positive group, most of the cells displaying green fluorescence were lymphocytes, and the fluorescence label appeared to be on the surface of the lymphocytes (Figure 1D). These results indicate that mAb 3C8 could detect lymphocytes and the binding sites for the mAb, which were localized on the surface of the cell.

### 2.2. Tissue Distribution of CD4-2-Positive Lymphocytes Analyzed by Flow Cytometry

To examine the tissue distribution of mAb (3C8)-positive CD4-2 cells, leukocytes isolated from the gill, liver, spleen, head-kidney, trunk-kidney, intestine and peripheral blood were analyzed by flow cytometry. Lymphoid cells and myeloid cells were observed within the leukocyte population based on the FSC (forward scatter) height and SSC (side scatter) height in the dot plot, and CD4-2-negative or -positive T cells in the different tissues detected by staining with the anti-CD4-2 mAb (3C8) (Figure 2A). A relatively high percentage of mAb 3C8-positive cells were present in the gill, spleen, head-kidney and trunk kidney, with 7.98 ± 0.89%, 11.09 ± 0.09%, 13.92 ± 0.53% and 13.45 ± 0.43% of the stained cells, respectively. However, lower percentages of mAb 3C8-positive cells were observed in the liver (4.29 ± 1.20%), intestine (1.78 ± 0.37%) and peripheral blood (4.53 ± 0.72%) (Figure 2B). In the spleen and kidney, the major lymphoid organs in fish, higher levels of 3C8-positive cells were present, suggesting that CD4-2 lymphocytes may be involved in the cellular immune response of olive flounder.

### 2.3. Expression of mRNA Profiles in CD4-2-Positive Lymphocytes

For further characterization of the identity of the anti-CD4-2 mAb (3C8)-positive cells, and the expression profiles of CD3ε, CD4-1, CD4-2, CD8α, CD8β, TCRα, TCRβ, IgL and IgM gene transcripts were examined in mAb 3C8-negative and -positive lymphocytes in the spleen and head-kidney leukocyte populations, sorted by FACS, and β-actin was used to normalize the mRNA expression levels. The leukocytes exhibited a characteristic distribution with regard to FSC area and SSC area in flow cytometry, and high levels of 3C8-positive populations resembling live lymphocyte-sized cells were measured in cell suspensions from the spleen and head-kidney (Appendix A). The results exhibited that CD4-2 transcripts were detected in the mAb 3C8-positive lymphocytes, whereas CD4-2 transcripts were not expressed in 3C8-negative lymphocytes. The CD3ε, CD4-1, TCRα and TCRβ transcripts were found in both mAb 3C8-negative and -positive lymphocytes, while the CD8α, CD8β, IgL and IgM transcripts were only expressed in mAb 3C8-negative lymphocytes (Figure 3). Gene transcripts of CD4-2-positive lymphocytes were only amplified by the primers expressing T cell transcripts and not B cell transcripts, indicating that the specificity of mAb 3C8 is specific to T cells. Moreover, mAb 3C8 seems to specifically recognize the CD4-2 lymphocytes in olive flounder T cells.

### 2.4. Two-Color Flow Cytometry to Confirm That the Staining of mAb 3C8 Was Specific for CD4-2 Lymphocytes in Olive Flounder

To show that the reactivity of mAb 3C8 was against only CD4-2 cells of olive flounder, and not against the CD4-1 molecule, we screened recombinant CD3ε, CD4-1 and CD-2 proteins and T cells with the specific mAbs we produced in previous studies against CD3ε and CD4-1 [1,14]. Anti-CD3ε mAb (4B2) and anti-CD4-1 mAb (10F8) detected 63 ng to 500 ng of recombinant CD3ε and CD4-1 protein, respectively. In addition, anti-CD3ε mAb (4B2) did not recognize recombinant CD4-1 and CD4-2 proteins, and anti-CD4-1 mAb (10F8) did not detect recombinant CD3ε and CD4-2 proteins (Appendix A). When the spleen, head-kidney and peripheral blood leukocytes were screened with the various mAbs by flow cytometry, the staining of CD3ε-positive T cells in the spleen, head-kidney and peripheral blood were ~6%, ~7% and ~2% of the total lymphocyte population, while the CD4-1-positive T cell population comprised ~7%, ~8% and ~2% of lymphocytes in the spleen, head-kidney and peripheral blood, respectively (Figure 4A). Two-color immunofluorescence analysis showed that a high percentage (~10%) of CD3ε and CD4-1 or CD4-2 double-positive cells were present in the population of head-kidney lymphocytes (Figure 4B), together with double-positive CD4-1 and CD4-2 lymphocytes (~15%), while a low percentage (~4%) of CD4-2 single-positive cells were present and only ~2% of CD4-1 single-positive cells were present (Figure 4C).

### 2.5. Response of CD4-1- and CD4-2-Positive Lymphocytes During an NNV Infection in Olive Flounder Determined by Flow Cytometry

Olive flounder infected with NNV showed clinical signs of disease, such as abnormal behavior and visual dysfunction associated with central nervous system damage, while no clinical symptoms were observed in the control groups injected with PBS. Tissues from infected fish were positive for NNV coat protein RNA, detected by PCR using specific primers [20]. The percentages of both CD4-1- and CD4-2-positive lymphocytes in fish infected with NNV were found to reach a peak and then gradually decreased until the end of the experiment, while there was no changes in the control group over the course of the infection. In both spleen and head-kidney cell populations, CD4-1 lymphocytes recognized by mAb 10F8 gradually increased and peaked at 14 days post infection (dpi) in spleen and at 7 dpi in head-kidney, respectively, and then started to decrease by 21 dpi (Figure 5). In the case of CD4-2-positive lymphocytes, the levels of these cells showed a slight decrease on 1 dpi, but levels rapidly increase at 3 dpi, and reached the peak at 7 dpi compared to the control group, and then their level decreased until the end of the experiment, in both spleen and head-kidney (Figure 6).

## 3. Discussion

The existence and function of helper T cells in teleost fish have been confirmed in a number of studies, suggesting that the presence of these T cell subsets is similar to those of higher vertebrates [21,22,23,24]. However, studies on the origin and primordial roles of helper T cell subsets in teleosts have been hindered by the lack of appropriate tools [1,16]. In previous studies, we produced mAbs against CD3ε and CD4-1 lymphocytes of olive flounder and confirmed their specificity and identity through Western blotting, flow cytometry and immunofluorescence staining [1,14]. In order to investigate the functions of CD4-2-positive T cells in teleosts, a monoclonal antibody against CD4-2 lymphocytes from olive flounder was produced in the present study. Through these three mAbs, we successfully characterized and established the distribution of CD4-2-positive lymphocytes in various immunological tissues of the fish, identified the lymphocyte subset co-expressing CD4-1 and CD4-2, and some lymphocytes were recognized that were uniquely expressing CD4-1 or CD4-2. We also detected lymphocytes co-expressing CD3ε and CD4-1, as well as CD3ε and CD4-2. Moreover, proliferation of CD4-1 and CD4-2 lymphocytes was observed when the fish were infected with NNV, suggesting that CD4-positive T cells are involved in a virus-specific immune response in olive flounder.

Although antibodies recognizing cell surface proteins have been successfully developed, these may exhibit different reactivities depending on the cell preparation used, e.g., fixed cell, live cell or cell lysates [13,25]. However, the anti-CD4-2 mAb (3C8) produced in this study showed a good level of responsiveness to the fixed cells used by immunofluorescence staining, the naïve form of CD4-2 expressed on the surface of the live cells used in flow cytometry, and the denatured cell lysate from the leukocytes used in Western blotting.

Leukocyte populations isolated from tissues contain not only lymphocytes but also granulocytes, monocytes and thrombocytes, with a similar size to lymphocytes under the microscope and in flow cytometry analysis [17,26]. Through previous studies, we revealed that the mAbs we produced specifically detect lymphocytes and did not target other cells of a similar size, using Giemsa staining and flow cytometry analysis [1,14]. The small cells recognized by mAb (3C8) were believed to be lymphocytes, because their dot plot profile in the flow cytometry was characteristic of lymphocytes [2,16,27,28], and very low populations of monocytes, macrophages and thrombocytes were present in the kidney and spleen [17,29,30].

In mammals, the lymphocytes are generated in the central lymphoid organs, including the thymus and bone marrow, whereas adaptive immune responses are initiated in the peripheral lymphoid tissues comprising the lymph nodes and spleen [31]. In teleosts, thymus, head-kidney and spleen are considered as the major lymphoid tissues, functioning as sites of antigen presentation and initiation of the adaptive immune responses [32,33,34,35]. Although the thymus is one of the most important lymphoid organs in teleosts, it can no longer be observed after around seven months in olive flounder [29]. Hence, in this study, the tissue distribution of the CD4-2 T lymphocytes was examined in various tissues, including the gill, liver, spleen, head-kidney, trunk-kidney, intestine and peripheral blood, instead of the thymus. The percentages of CD4-2-positive lymphocytes were high in systemic lymphoid organs (spleen, ~11%; head-kidney, ~14%) and low in non-lymphoid tissues (liver, ~4%; intestine, ~2%; peripheral blood, ~5%). These results support previous findings showing that the spleen and head-kidney to be the major lymphoid organs related to the adaptive immune response in teleosts [31,32]. Moreover, the tissues specific for CD4-2-positive populations were consistent with those observed for other teleost species [5,9,10,22,36].

The expression of lymphoid cell marker genes in cell populations that reacted with mAb (3C8) was analyzed by RT-PCR. Based on the transcript pattern, CD4-2-positive lymphocytes express transcripts for CD3ε, CD4-1, CD4-2, TCRα and TCRβ. On the other hand, CD4-2-negative cells expressed CD3ε, CD4-1, TCRα, TCRβ, CD8α, CD8β, IgL and IgM transcripts. These results implied that the strong expression pattern of the CD8, IgL and IgM transcripts observed in the CD4-2-negative cells is distinctive in the CD4-negative T cell subsets in teleosts [25,33,34]. Additionally, these results revealed that the expression of CD4-2 lymphocytes is associated with cell surface expression of CD3ε and TCR, and CD4-2 lymphocytes are able to co-express with CD4-1 lymphocytes, as described in many previous studies [11,17,37].

The proportion of CD3ε T cells was ~69% in humans and ~40% in salmonids [11,38], while the percentage of CD3ε T lymphocyte in head-kidney was detected at only ~6% in this study and ~3% in our previous study [14]. It was thought that the presence of the short length of the CD3ε chain in the extracellular domain in olive flounder may influence the low population of CD3ε lymphocytes, and the difference in population in the previous and current studies was due to various conditions, including fish status, the developmental stage of the fish, isolation of leukocytes, and the use of antibodies [11,14]. Nonetheless, the double labeling of lymphocytes in head-kidney using anti-CD3ε and anti-CD4 mAbs in flow cytometry showed the presence of CD3ε+/CD4-1+ cells (~10%) as well as CD3ε+/CD4-2+ cells (~11%), and this might be because CD3ε T cells containing CD4+ and CD8+ cells were detected, corresponding to previous studies for several other fish species [11,39,40,41].

In addition, it was expected that three distinct CD4 lymphocyte populations (e.g., CD4-1 single positive, CD4-2 single positive as well as CD4-1 and CD4-2 double-positive lymphocytes) were recognized in olive flounder because the predominant lymphocyte subset co-expressing both CD4-1 and CD4-2, and minor lymphocyte population expressing either CD4-1 or CD4-2, have also been reported in several teleost species [10,37,40], as well as in mammals [36,42]. Until now, CD4 T cell subsets could not be compared in olive flounder because of the lack of appropriate mAbs to detect both the CD4-1 and CD4-2 molecules in this species, reflecting a similar situation for other teleost species [37]. In this study, slight differences were observed in the level of CD4-1 and CD4-2 cell populations in the spleen (CD4-1, 7%; CD4-2, 11%) and head-kidney (CD4-1, 8%; CD4-2, 14%), and three kinds of CD4 lymphocytes (CD4-1+/CD4-2+, ~14%; CD4-1+/CD4-2-, ~2%; and CD4-1-/CD4-2+, 4%) were expressed in olive flounder. Taken together, the results in the present study suggested the CD4 T cell subsets are composed of concurrent or unique subpopulations of the CD4-1 and CD4-2 molecules.

T cells expressing the CD4 surface protein are equivalent to helper T cell in mammals, and play an important role in humoral immune response against bacterial and viral infection [3]. Although the variation of CD4 lymphocyte against antigen stimulation was reported in several studies, the proliferation of CD4 lymphocytes during NNV infection is largely unclear. NNV, which induces the induction of genes encoding the B and T lymphocyte markers, triggers Th1-induced cell-mediated immunity in olive flounder [3,21,43,44]. In the present study, proliferation of both CD4-1 and CD4-2 lymphocytes were observed in fish after infection with NNV. The results, which showed significant increases in both CD4-1 and CD4-2 lymphocytes at 7 dpi in head-kidney, were parallel with the previous observations indicating that the cell-mediated responses against viruses are generally observed at 5 to 7 days after viral infection [45]. Interestingly, the proliferation of the CD4-2-positive cells in the spleen appeared earlier than that of the CD4-1-positive cells, and the increase in the CD4-2-positive cells was also higher than that of the CD4-1-positive cells. These data suggested that the CD4-2-positive cells in flounder could proliferate during viral infection in which the Th1 immune responses are predominantly induced. Notably, CD4-2 lymphocytes, rather than CD4-1 lymphocytes, could have a primary role in Th1 immune responses. However, further comparison with CD8 lymphocytes and determination of specific cytokines are required to fully elucidate the overall role of the Th cell functions in the fish’s immune response.

To date, many bacterial vaccines have been developed for teleosts, but only few viral vaccines have been commercialized. This is because it is difficult to develop and produce viral vaccines for fish, and not always easy to select the appropriate virus candidates for the vaccine. Therefore, the measures to effectively prevent the viral diseases are crucial. Biomarker research is expected to play an important role in the development of new vaccines [46,47]. It has been shown that the viral vaccines are able to induce a cellular immune response, including CD4 and CD8 lymphocytes, against viral infections [48]. In human vaccinology, the response of biomarkers is used to help develop vaccines and evaluate vaccine efficacy. Thus, the mAb produced in this study, indicating upregulation of T cell responses in olive flounder viral infections, could be used to help in the design of viral vaccines.

To summarize our findings, the anti-CD4-2 mAb (3C8) newly developed in the present study demonstrated that the functional characteristics of the CD4-2 T cells in olive flounder are similar to those of mammals in terms of their cell morphology, tissue distribution and gene expression. Using this mAb, the subpopulation of CD4 T cell subsets and the variations of these T cell populations were able to explain the helper T cell function for the CD4-positive T cells during viral infection, suggesting the involvement of CD4 T cells in the anti-viral immune response. Besides, it was demonstrated that the CD4-2 T cells increase more and earlier than the CD4-1 T cells during viral infection. This suggests that the CD4-2 lymphocytes might have a main role in the immune response of olive flounder against viral infections. Therefore, our mAbs against CD4-1 and CD4-2 T lymphocytes could be useful tools for understanding and expanding our knowledge on Th cell function in teleosts.

## 4. Materials and Methods

### 4.1. Expression and Purification of the Recombinant CD4-2 Antigen

The partial CD4-2 gene sequence (507 bp; 58–564) (GenBank accession no. AB640684.1, https://www.ncbi.nlm.nih.gov/nuccore/AB640684.1), located in the extracellular domain, was amplified with primers flanked by restriction endonuclease Xba I and Not I (Forward : 5′- TATATCTAGACGTGATCCTAACAAAACCCAGGC -3′, Reverse : 5′- GGTGGCGGCCGCCTAGTGGTGGTGATGGTGGTGGTGGTGAGCAGGTTCTTCAACTTTGATCTT -3′), and the resulting amplicon was inserted into the same enzyme sites of the pLexsy-Sat2.1 plasmid (Jena Bioscience, Jena, Germany) of an *Leishmania* expression system. The underlined sequences indicate the restriction enzyme sites associated with the primer’s name. Selected colonies confirmed by DNA sequencing analysis were inoculated into Luria-Bertani (LB) broth (Merck, Darmstadt, Germany) containing 100 µg/mL of ampicillin. A total of 10 µg of the plasmid obtained, containing the CD4-2 gene sequence, was digested with Swa I, and transfected to *Leishmania tarentolae (L. tarentolae)* by electroporation. The protein was purified using a Ni-NTA affinity chromatography column (Elpisbio, Daejeon, Korea). To eliminate non-specific proteins, the column was washed with 1x phosphate buffered saline (PBS) containing 10 mM imidazole, and the specific protein was eluted with 1x PBS containing 500 mM imidazole. The correct identity of the eluted protein was confirmed by running it on a 15% SDS-PAGE gel under reducing conditions, stained with Coomassie blue and the protein band at approximately 20 kDa was subjected to MALDI–TOF/TOF MS analysis as previously described [1].

### 4.2. Production of mAb (3C8) Specific to CD4-2 Lymphocytes from Olive Flounder

The purified recombinant CD4-2 antigen was used to immunize three 6-week-old female BALB/c mice. The antigen (150 µg) was mixed with Freund’s complete adjuvant (FCA) (1:1 *v/v*) for the first round of the immunization, while for the second and third immunization the antigen was mixed with Freund’s incomplete adjuvant (FIA) (1:1 *v/v*), all delivered by intraperitoneal injection. Two weeks after the last injection, a final booster immunization was performed injecting 10 µg of unadjuvanted antigen into the tail vein of the mice. Three days after this booster immunization, spleen cells were harvested from the mice and fused with Sp2/o myeloma cells using polyethylene glycol according to standard protocols [49]. The hybridomas were grown in 96-well plates on a feeder layer of mouse blood cells. An enzyme-linked immunosorbent assay (ELISA) and Western blotting were used to select positive hybridomas. The clone that showed the best specificity for the olive flounder CD4-2 antigen was mAb 3C8, and was therefore selected for further analysis. The isotyping of the mAb was performed by an ELISA using mouse monoclonal antibody isotyping reagents (Sigma-Aldrich, St.Louis, Missouri, USA).

### 4.3. Preparation of Olive Flounder Leukocytes

Healthy olive flounders, 10–20 cm in size (Samjin fish farm, Namhae, Korea), were bled from the caudal vessel using heparinized syringes after anesthetizing the fish by immersion in 0.1 g/L of ethyl 3-aminobenzoate methane-sulfonic acid (Sigma-Aldrich, St.Louis, Missouri, USA). Blood samples were immediately diluted 1:4 in cold Dulbecco’s Modified Eagle’s medium (DMEM) (Thermo Fisher Scientific, Waltham, Massachusetts, USA) containing heparin. Other organs, including the gills, liver, spleen, head-kidney, trunk-kidney and intestine, were homogenized and filtered separately through a cell strainer (BD Falcon, Bedford, Massachusetts, USA), and the leukocytes were isolated using a Percoll (GE Healthcare, Chicago, Illinois, USA) gradient [14]. All procedures involving cells were carried out at 4 °C under sterile conditions. Cell concentration and viability were determined using a hemocytometer.

### 4.4. Transfection

To generate pKIN/CD3ε, pKIN/CD4-1, and pKIN/CD4-2, the partial CD3ε, CD4-1, and CD4-2 DNA fragments flanked by two Sfi I sites (CD3ε F : 5′-TATAGGCCACCGGGGCCATGAAATTTACATCACTGTTGC-3′, CD3ε R : 5′-GGTGGGCCCCAGAGGCCTTGGTAGGTAGGTTGAGCTCGAT-3′, CD4-1 F : 5′-TATAGGCCACCGGGGCCATGGATCCCAGAGGAGAGATAATGAATG-3′, CD4-1 R : 5′-GGTGGGCCCCAGAGGCCCACGTAGTCTCCTCCGTC-3′, CD4-2 F : 5′-TATAGGCCACCGGGGCCGTGATCCTAACAAAACCCAGGC-3′, CD4-2 R : 5′-GGTGGGCCCCAGAGGCCAGCAGGTTCTTCAACTTTGATCTT-3′) were PCR amplified and inserted into the two Sfi I sites of the p514 vector (produced in our lab) containing a red fluorescent protein sequence [50]. The constructed plasmids were purified using DNA Spin miniprep kits (Intron, Sungnam, Korea) and quantified using a NanoDrop spectrophotometer. For transfection, HEK 293F cells were seeded into 8-well chamber slides or 24-well plates, grown to a 90% confluence [51], and transfected with the plasmids using Lipofectamine2000 (Invitrogen, Carlsbad, California, USA) according to the manufacturer’s instructions. After 4 h, the transfectants were transferred to Dulbecco’s Modified Eagle’s medium (DMEM) (Thermo Fisher Scientific, Waltham, Massachusetts, USA) containing 2% fetal bovine serum (FBS). After 48 h, cells were used for immunofluorescence staining and flow cytometry analysis as described below.

### 4.5. Western Blotting

Different amounts of recombinant CD3ε, CD4-1, and CD4-2 protein samples and 1 × 10^7^ cells/mL of cell suspensions from the spleen, head-kidney and peripheral blood resuspended in RIPA buffer were separated on 15% SDS-PAGE under reducing conditions (80 V for 20 min, 120 V for 90 min) [1,12]. Protein bands in gels were transferred to methanol-activated polyvinylidene fluoride (PVDF) membranes at 50 mA for 90 min. The membranes were blocked with 5% (*w/v*) skim milk in 1x PBS containing 0.1% (v/v) Tween 20, and then incubated with specific monoclonal antibodies (CD3ε, 4B2; CD4-1, 10F8; CD4-2, 3C8) followed by HRP-conjugated goat anti-mouse IgG (Thermo Fisher Scientific, Massachusetts, USA) [1,12]. These membranes were then stained with a SuperSignal West Pico Chemiluminescent Substrate kit (Thermo Fisher Scientific, Massachusetts, USA).

### 4.6. Flow Cytometry

A total of 1 × 10^7^ leukocytes per sample from the gill, liver, spleen, head-kidney, trunk-kidney, intestine and peripheral blood were isolated as described above and centrifugated at 500 *g* for 3 min. Prior to incubation with mAb, cells were blocked with 0.1% bovine serum albumin (BSA) in 1× PBS for 30 min. Leukocytes were then treated with mAb 3C8, followed by FITC-conjugated AffiniPure goat anti-mouse IgG (Jackson ImmunoResearch, West Grove, Pennsylvania, USA) for 1 h. Two-color flow cytometry analysis was conducted against cell surfaces between CD3ε and CD4-1, between CD3ε and CD4-2, and between CD4-1 and CD4-2. For the analysis of cell surface between CD3ε and CD4-1, leukocytes from tissues were first incubated with mAb 10F8 (anti-flounder CD4-1 mouse IgG2) followed by PE-conjugated goat anti-mouse IgG2 for 1 h. Leukocytes were washed with 1× PBS 3 times and then reacted with mAb 4B2 (anti-flounder CD3ε mouse IgG1) followed by FITC-conjugated AffiniPure goat anti-mouse IgG1. The analysis of cell surface CD3ε and CD4-2 was performed similar way. Leukocytes were stained with mAb 3C8 (anti-flounder CD4-2 mouse IgG2b) followed by PE-conjugated goat anti-mouse IgG2. After washing, leukocytes were treated with mAb 4B2 followed by FITC-conjugated AffiniPure goat anti-mouse IgG1. For double staining with CD4-1 and CD4-2, leukocytes were first incubated with mAb 3C8 after biotinylated with NHS 648 ester (BioActs, Incheon, Korea) that was able to show red fluorescence. Cells were washed and then reacted with mAb 10F8 followed by FITC-conjugated AffiniPure goat anti-mouse IgG2. Cells binding with mAbs were analyzed by a FACSCalibur^TM^ (BD biosciences, Bedford, Massachusetts, USA). At least 30,000 events were measured for each sample.

### 4.7. Immunofluorescence Staining

The CD4-2-positive HEK 293F cells were fixed onto 8-well chamber slides with 4% paraformaldehyde (Intron, Sungnam, Korea) for 15 min. A final concentration of 1 × 10^5^ cells from the head-kidney were prepared on a slide glass using a cytological centrifuge (Hanil Science Industrial, Gimpo, Korea) at 30 *g* for 5 min. After centrifugation, the cells were fixed with 4% paraformaldehyde for 15 min, blocked with 0.1% BSA in 1× PBS for 30 min, and stained with anti-CD4-2 mAb (3C8) for 1 h, followed by FITC-conjugated AffiniPure goat anti-mouse IgG for 1 h. Negative controls were only stained with FITC, and three washes with 1× PBS were carried out between each step. Cells were then stained with DAPI for 10 min at room temperature. HEK293F cells and leukocytes recognized by mAb (3C8) were examined under a fluorescence microscope, Olympus FV 1000 (Olympus, Seoul, Korea).

### 4.8. RT-PCR with Flow Cytometry Sorted Leukocytes

Leukocytes (1 × 10^6^ cells/mL in 1× PBS) from the spleen and head-kidney were prepared and stained as described in the flow cytometry section and sorted using a FACSARIA III cell sorter (BD Biosciences, San Jose, USA). Lymphocytes from the spleen and head-kidney were separated into two groups: 3C8-positive and -negative cells. Total RNA was extracted from 30,000 sorted cells of each population using an easy-BLUE Total RNA Extraction Kit (Intron, Sungnam, Korea) and reverse transcribed into cDNA using a TOPscript cDNA Synthesis Kit with Oligo (dT) primers (Enzynomics, Daejeon, Korea) according to the manufacturer’s instructions. Specific primers, including CD3ε, CD4-1, CD4-2, CD8α, CD8β, TCRα, TCRβ, IgL, IgM and β-actin were used for the RT-PCR and are shown in Table 1. For the RT-PCR, 1 μL of cDNA template and 10 pM of each primer were used together with an AccuPower ProFi Taq PCR premix (Bioneer, Daejeon, Korea). The PCR conditions were as follows: one cycle of 95 °C for 3 min, 34–40 cycles at 95 °C for 20 s, 55–65 °C as the annealing temperature for 20 s, and 72 °C for 50 s. The PCR products on a 1% agarose gel were stained with RedSafe nucleic staining solution (Intron, Sungnam, Korea). Images were visualized by an AE-9000E-graph (ATTO Corporation, Tokyo, Japan). Each analysis was repeated three times.

### 4.9. Assessing Populations of CD4-1- and CD4-2-Positive Lymphocytes in Olive Flounder Infected with NNV

Olive flounders (Samjin fish farm, Namhae, Korea), 10–20 cm in size, were placed in three replicate tanks each containing 100 fish. The fish were acclimated in laboratory conditions for 2 weeks and fed with dry food pellets once a day. Fish were artificially infected with 100 µL of 1 × 10^7^ TCID50/mL NNV particles intramuscularly or NNV particles mixed with Freund’s complete adjuvant (FCA) at an equal volume, and the negative controls were injected with same volume of 1× PBS. Three tanks were maintained at 20–25 °C. At 0, 1, 3, 7, 14 and 21 days post-infection (dpi), the spleen and head-kidney were removed from five fish in each tank. After isolating the leukocytes, cells were incubated with 10F8 (anti-CD4-1 mAb) or 3C8 (anti-CD4-2 mAb), followed by FITC-conjugated AffiniPure goat anti-mouse IgG. The CD4-1- and CD4-2-positive cell populations were compared to the negative controls stained with only FITC-conjugated AffiniPure goat anti-mouse IgG.

### 4.10. Statistical Analysis

The data are displayed as the mean ± standard deviation (SD). Statistical analysis was performed using one-way analysis of variance (ANOVA) using the GraphPad Prism v7.0 software.

## Figures and Tables

**Figure 1 ijms-21-04180-f001:**
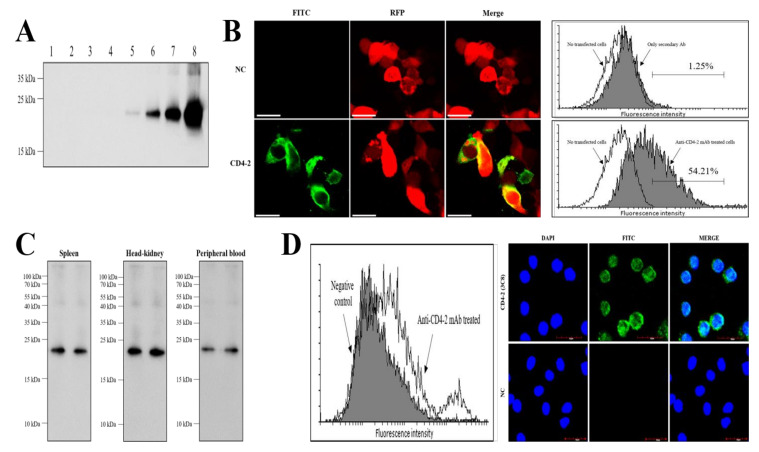
Identification of the anti-CD4-2 monoclonal antibody (3C8). (**A**) The antibody was screened for specificity and sensitivity against increasing amounts of recombinant CD4-2. Lane 1, rCD3ε at 500 ng; Lane 2, rCD4-1 at 500 ng; Lane 3, rCD4-2 at 16 ng; Lane 4, rCD4-2 at 32 ng; Lane 5, rCD4-2 at 63 ng; Lane 6, rCD4-2 at 125 ng; Lane 7, rCD4-2 at 250 ng; Lane 8, rCD4-2 at 500 ng. (**B**) Reactivity of anti-CD4-2 mAb to transfected 293F cells stably expressing flounder CD4-2 molecules. Scale bar = 20 um. (**C**) Western blotting analysis of leukocyte cell lysates from spleen, head-kidney and peripheral blood of olive flounder using anti-CD4-2 mAb (3C8), followed by HRP-conjugated goat anti-mouse IgG. (**D**) Flow cytometry analysis of head-kidney leukocytes stained with anti-CD4-2-mAb. Negative control; shaded area, Anti-CD4-2-mAb treated; black line. Immunofluorescence staining of head-kidney leukocytes incubated with anti-CD4-2 mAb (3C8), followed by FITC-conjugated goat anti-mouse IgG. Immuno-reactivity appears green.

**Figure 2 ijms-21-04180-f002:**
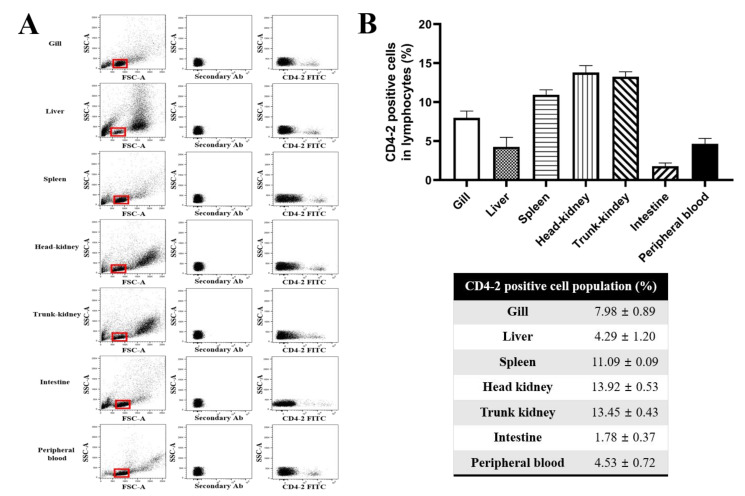
Staining pattern of mAb 3C8 in leukocytes in different tissues analyzed by flow cytometry. (**A**) Gating strategy for flow cytometry analysis. Lymphocytes were gated on an FSC (forward scatter) and SSC (side scatter) dot blot represented by a red box, shown in the left panel; cells treated with only FITC (secondary Ab) were represented in the middle panel; cells incubated with anti-CD4-2 mAb, followed by FITC were represented in the right panel. (**B**) Percentages of CD4-2-positive lymphocytes in different tissues, represented as the mean ± SD of at least five fish.

**Figure 3 ijms-21-04180-f003:**
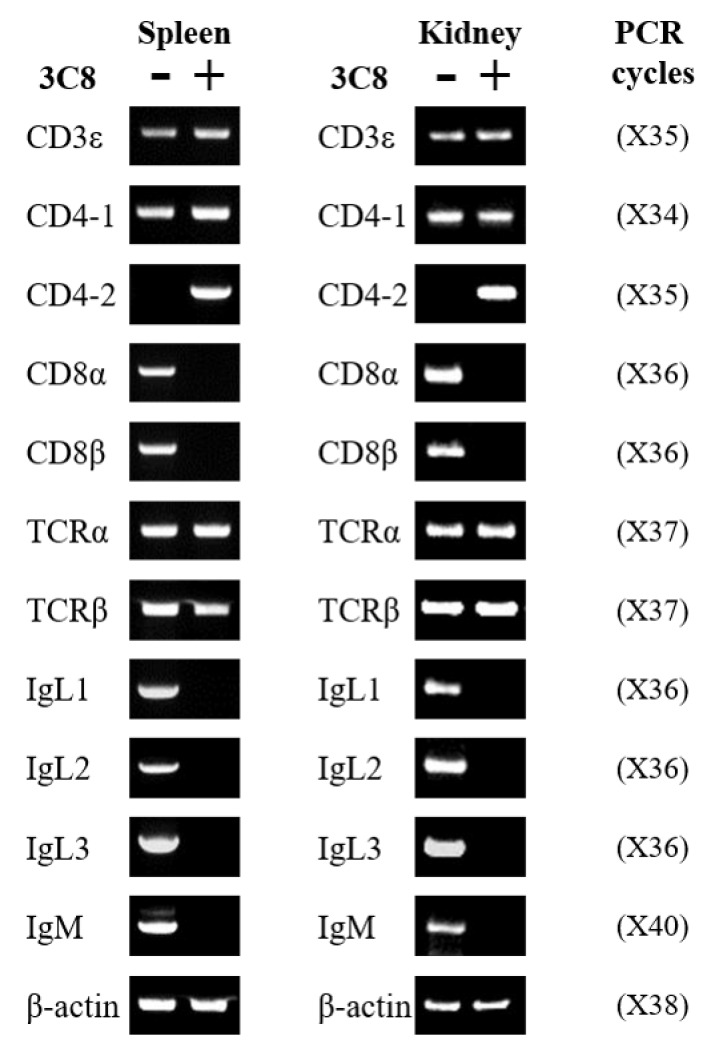
Gene expression profiles of the cell marker genes in flow-sorted spleen lymphocytes and head-kidney lymphocytes. Specific oligonucleotide primers for T cell and B cell gene expression were employed for RT-PCR and β-actin was used as the housekeeping gene. Numbers in parenthesis indicate PCR cycles. -; mAb 3C8-negative lymphocytes, +; mAb 3C8-positive lymphocytes.

**Figure 4 ijms-21-04180-f004:**
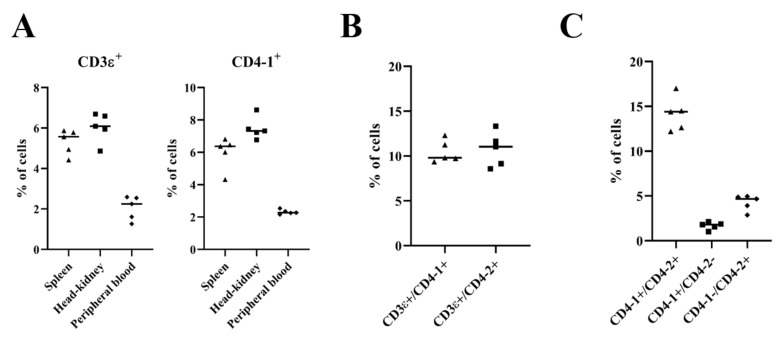
Two-color immunofluorescence staining results of CD3ε^+^/CD4-1^+^ and CD3ε^+^/CD4-2^+^ T lymphocytes, and CD4-1^+^/CD4-2^+^, CD4^-^1^+^/CD4-2^-^ and CD4-1^-^/CD4-2^+^ T lymphocytes in head-kidney leukocytes. (**A**) Percentage of CD3ε and CD4-1 positive cells in the spleen, head-kidney and peripheral blood analyzed by flow cytometry. (**B**) Percentage of CD3ε^+^/CD4-1^+^ and CD3ε^+^/CD4-2^+^ cells in head-kidney. (**C**) Percentage of CD4-1^+^/CD4-2^+^, CD4-1^+^/CD4-2^-^ and CD4-1^-^/CD4-2^+^ cells in head-kidney. Data are represented as the mean ± SEM of positive cells over the total lymphocytes of at least five fish.

**Figure 5 ijms-21-04180-f005:**
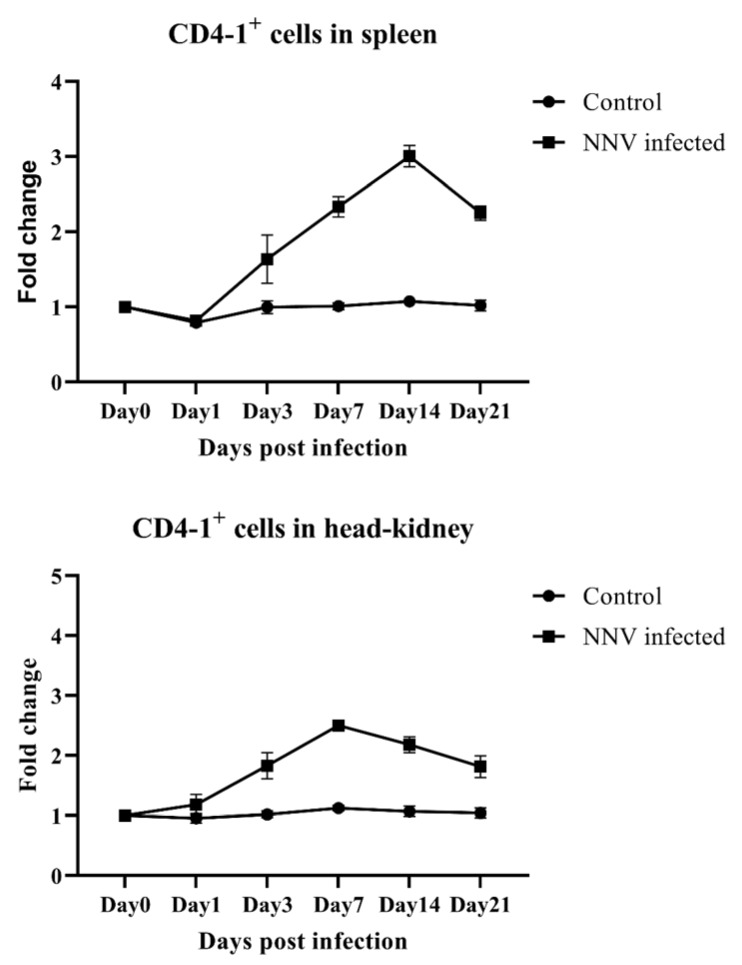
In vivo proliferation of CD4-1-positive lymphocytes in response to a nervous necrosis virus (NNV) infection analyzed by flow cytometry. Leukocytes from the spleen and head-kidney were isolated at 0, 1, 3, 7, 14 and 21 days post infection, and stained with mAb 10F8, followed by FITC-conjugated goat anti-mouse IgG. Lymphocytes were gated on an FSC and SSC dot plot and analyzed by flow cytometry. The negative control was incubated without anti-CD4-1 mAb (10F8). Data are represented as the mean ± SD of three analyses.

**Figure 6 ijms-21-04180-f006:**
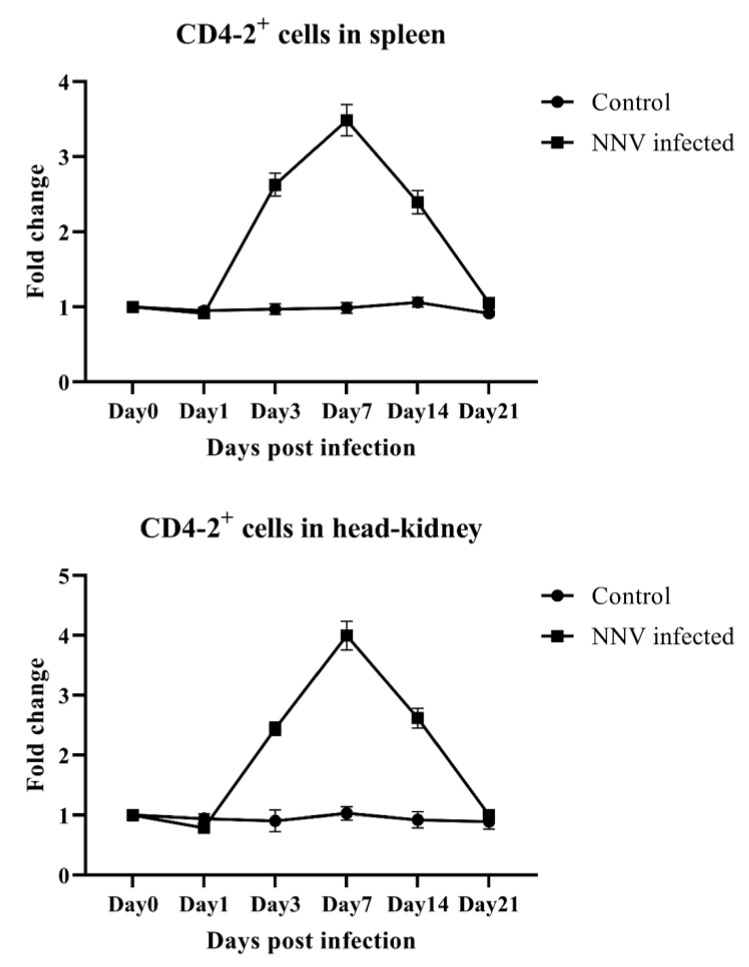
Variations in CD4-2-positive lymphocyte percentage in the spleen and head-kidney after injection with a nervous necrosis virus (NNV). Leukocytes from the spleen and head-kidney were obtained at 0, 1, 3, 7, 14 and 21 days post infection, and incubated with anti-CD4-2 mAb (3C8), followed by FITC-conjugated goat anti-mouse IgG. Data are presented as the mean ± SD of three analyses.

**Table 1 ijms-21-04180-t001:** Oligonucleotide primer sequences used for the RT-PCR analysis.

Primer Name		Sequence (5′–3′)
CD3ε (AB081751.1)	Forward	ATGAAAATCAACACCATGGATGTC
	Reverse	TCCCGTCCTGTTCACAATAGA
CD4-1 (AB643634.1)	Forward	ATGAATCCCAGAGGAGAGATAATG
	Reverse	CACGTAGTCTCCTCCGTCTTC
CD4-2 (AB640684.1)	Forward	GTGATCCTAACAAAACCCAGGCAG
	Reverse	AGCAGGTTCTTCAACTTTGATCTT
CD8α (AB082957.1)	Forward	ATGGACCAAAAGTGGATTCAGATG
	Reverse	AACATGTGTGTTGTTCTTCATCTG
CD8β (AB643633.1)	Forward	ATGAACCCGCTGCCGCTG
	Reverse	GGGCATCTGTCTCATCTTCTG
TCRα (AB053227.1)	Forward	ATGCTCTCACTGCATCTTGGT
	Reverse	GACTCTGTGACTGAGCCACAG
TCRβ (AB053228.1)	Forward	ATGATTCCAAGCCTCAACACC
	Reverse	GTGGTTCTGCTTCTCAGCTGA
IgL1 (AB819734.1)	Forward	ATGAGCTTTACCTCCGTCCTC
	Reverse	GGACTGGGAACACTGGTCTCT
IgL2 (AB819735.1)	Forward	ATGATGGTTTTTCTGAGTCAGGAG
	Reverse	CTCCGAGCAGCGGTCAGG
IgL3 (AB819736.1)	Forward	ATGCTGGGGACCCTCTGC
	Reverse	GTGGTACAGACGGACTTGTTG
IgM (AB052744.1)	Forward	ATGTTTCCTGTAGCTGTGCTG
	Reverse	CTGGGCCTTGCATGGTATGTT
β-actin (HQ386788.1)	Forward	ATGGAAGATGAAATCGCCGCA
	Reverse	GAAGCATTTGCGGTGGACGAT

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
