# Peer review of "Characterization of CD4-Positive Lymphocytes in the Antiviral Response of Olive Flounder (Paralichthys oliveceus) to Nervous Necrosis Virus"

_ijms, 2020, doi:10.3390/ijms21114180_

Round 1
Reviewer 1 Report
The authors report the production of a monoclonal antibody against CD4-2 T lymphocytes from olive flounder, which they used (in addition to monoclonal antibodies previously developed against CD3ε and CD4-1 lymphocytes) to characterize the CD4 T lymphocytes in olive flounder. Their findings show that these cells are equivalent to the mammalian T helper cells and that the CD4-2 T lymphocytes play an important role in the Th1 immune response against viral infections in olive flounder.
While the finding that the CD4 T lymphocytes in teleost fish are equivalent to the mammalian T helper cells is not new, the application of the monoclonal antibody developed to in vivo studies in this manuscript warrant its publication as it extends the number of fish species in which CD4 T lymphocytes have been characterized.
The authors explain why the thymus was not included in this study. To keep the manuscript current it would have been useful to also comment on the salmon bursa; i.e., does olive flounder also have this structure that consists mostly of T lymphocytes?
A minor comment: The manuscript needs to be reviewed for English grammar.
Author Response
We greatly appreciate the editor and the reviewers for giving us the chance to improve our manuscript. We are especially thankful for the in-depth and valuable comments from the reviewers. The summary of our changes is outlined in the following: the reviewers’ comments are underlined and our responses are in bold letters. The edited contents and sentences were highlighted in yellow in manuscript and our response to reviewer’s comments is below. We hope the reviewers find our efforts to be satisfactory. We also addressed the comments by the reviewers below:
Response to reviewer’s comments
#Reviewer 1
Comments and Suggestions for Authors
The authors report the production of a monoclonal antibody against CD4-2 T lymphocytes from olive flounder, which they used (in addition to monoclonal antibodies previously developed against CD3ε and CD4-1 lymphocytes) to characterize the CD4 T lymphocytes in olive flounder. Their findings show that these cells are equivalent to the mammalian T helper cells and that the CD4-2 T lymphocytes play an important role in the Th1 immune response against viral infections in olive flounder.
While the finding that the CD4 T lymphocytes in teleost fish are equivalent to the mammalian T helper cells is not new, the application of the monoclonal antibody developed to in vivo studies in this manuscript warrant its publication as it extends the number of fish species in which CD4 T lymphocytes have been characterized.
The authors explain why the thymus was not included in this study. To keep the manuscript current it would have been useful to also comment on the salmon bursa; i.e., does olive flounder also have this structure that consists mostly of T lymphocytes?
Answer : Thank you for your comments.
The thymus, one of the most important lymphoid organs in teleost, is located near the gill cavity and present even in adult fish, although the volume diminishes with age or sexual maturation. In olive flounder, the thymus was identified in about 60% of the fish at 3 and 4 months. Only 20% of fish still had a thymus at 6 months. After that, the thymus disappeared gradually [Liu, Y.; Zhang, S.; Jiang, G.; Yang, D.; Lian, J.; Yang, Y., The development of the lymphoid organs of flounder, Paralichthys olivaceus, from hatching to 13 months. Fish & shellfish immunology 2004, 16 (5), 621-632.].
In discussion, we have explained why the thymus was not included in this study. “The thymus can no longer be observed after around 7 months in olive flounder. Hence, in this study, the tissue distribution of the CD4-2 T lymphocytes was examined in various tissues including gill, liver, spleen, head-kidney, trunk-kidney, intestine and peripheral blood, instead of the thymus.”
Furthermore, CD4 molecules have been identified, CD4-1 and CD4-2 lymphocytes, in atlantic salmon (Salmo salar) [Moore, L. J., et al. "CD4 homologues in Atlantic salmon." Fish & shellfish immunology 26.1 (2009): 10-18.]. Thus, we have added this paper in references. “In teleosts two types of CD4-like molecules, CD4-1 and CD4-2, have been reported in fugu (Takifugu rubripes), rainbow trout (Oncorhynchus mykiss), carp (Cyprius carpio), sea bass (Dicentrarchus labrax) and atlantic salmon (Salmo salar) [6 – 11].”
A minor comment: The manuscript needs to be reviewed for English grammar.
Answer : We appreciate the reviewer’s comment. We have made the English grammar corrections as the reviewer suggested.
Reviewer 2 Report
The manuscripts n. JIMS-800708 deals with the characterization of CD4-2 putative T lymphocytes in Japanese(olive) Flounder in normal condition and their distribution after an infection disease (NNV).
The authors have also cited in the text a very manuscript: Mol Immunol. 2017 May;85:155-165. doi: 10.1016/j.molimm.2017.02.015. Epub 2017 Mar 4. From the MDPI that is similar: Characterizations of CD4-1, CD4-2 and CD8β T cell subpopulations in peripheral blood leukocytes, spleen and head kidney of Japanese flounder (Paralichthys olivaceus). Xing J, Ma J, Tang X, Sheng X, Zhan W. Reading this manuscript, a few differences are observed. Something of innovative sound could be the functional characterization of T- lymphocyte in J.flounder, but there are already several manuscripts that have defined that functionality in other species of fish (Biology (MDPI). 2015 Dec; 4(4): 640–663, Published online 2015 Sep 25. doi: 10.3390/biology4040640;(IJMS)https://www.mdpi.com/1422-0067/21/7/2439/htm; The Journal of Immunology 177(6):3939-51
DOI: 10.4049/jimmunol.177.6.3939 activation of lymphocytes: Front. Immunol., 31 October 2019 | https://doi.org/10.3389/fimmu.2019.02579). The distribution of lymphocytes during infection in fish is not new (https://journals.plos.org/plosone/article?id=10.1371/journal.pone.0147477; Microorganisms. 2019 Dec; 7(12): 627; Novel Technologies for Vaccine Development
Igor S Lukashevich, Haval Shirwan eds.; Vaccine 31 (2013) 1224–1230; 2019- doi: 10.3390/microorganisms7120627; https://www.jimmunol.org/content/200/1_Supplement/99.4).
Thus, the research interest in the field is not really high in the field of fish immunologists when the manuscript is written in this mode. I can suggest to transform and re-write the manuscript by applying the data in exploring a vaccination program. It is possible to use "vaccination " in silico with “bit-fish” to predict the behavior of the immunological system in presence of one of two pathogens (https://academic.oup.com/bioinformatics/article/33/19/3065/3854934).
Introduction:
Line 3: please insert also the reference of sea bass (already present in the manuscript, n.33).
Lines 4-8: the description is redundant, please synthesize the content.
Lines 14-23: the assumption does not take into account the manuscript cited in n.22 and insert in the general comment. A wide characterization of CD4 typology and CD8 has been done in 2017 for Japanese Flounder.
Results:
Fig.3: The authors should summarize the result in the figure (i.e. insert only the flowcytometry graphic and translate the particulars of western-blot and gates in supplementary materials) because the manuscript is heavy to read.
Fig.4: The authors should translate fig.4 A in supplementary material and leave the information of the positivity in the text.
NNV infection- from lines 12 pag.15- lines 1-5 pag.16 (Fig.5). This part is the core of the work and needs implementation by information about the diffusion of the pathology in the organs.
Discussion
The discussion should be rewritten. The authors did not insert a real discussion from their data and the n.22 reference that is mainly a previous copy of their work. In fact, they merely discuss (page 9, lines 5-7): “The small cells recognized by mAb (3C8) were believed to be lymphocytes, because the very low population of monocytes, macrophages and thrombocytes were present in kidney and spleen [22, 23]. Whereas, many things may be discussed in comparison with the data published by other authors.
Round 2
Reviewer 2 Report
The authors have improved the manuscript. Anyway, they lack image support for their important affirmations relative to the specific positivity of their developed mAb 3C8:
"Through previous studies, we revealed that the
mAbs we produced specifically detect lymphocytes and did not target other cells of a similar size, using Giemsa staining and flow cytometry analysis [1, 14]. The small cells recognized by mAb (3C8) were believed to be lymphocytes because their dot plot profile in flow cytometry was characteristic of lymphocytes [2, 16, 23, 24], and very low
population of monocytes, macrophages and thrombocytes were present in kidney and spleen [17, 25, 26]."
I wish to see immunohistochemistry on tissues (not only on "cells" to be sure that the affirmation by authors that 3C8 mAB selectively marked lymphocytes. IS TOO simplistic wrote "there are few granulocytes, macrophages thrombocytes in spleen and head kidney". On the contrary, in my experience, these organs are specialized as primary organs and active producers for these cells from larvae to adulthood. Moreover, thrombocytes have similar size of lymphocytes. Thus, I wish to have proof of their "sparse" presence in the organs. It is necessary to VALIDATE ALL EXPERIMENTS AND the conclusions!. They could insert this information (a table with: blanc on thymus omitting 3C8 and only II antibody, immunohistochemistry with 3C8 on 1) Thymus; 2) head kidney; 3)spleen; 4) gills, 4)intestinal tract.
I cannot approve of this work without this information.
